# Domino-like multi-emissions across red and near infrared from solid-state 2-/2,6-aryl substituted BODIPY dyes

Dan Tian[1], Fen Qi[1], Huili Ma[1], Xiaoqing Wang[2], Yue Pan[1], Runfeng Chen [3],
Zhen Shen[4], Zhipeng Liu [1], Ling Huang [1] & Wei Huang[1,3]

Considerable achievements on multiple emission capabilities and tunable wavelengths have been obtained in inorganic luminescent materials. However, the development of organic counterparts remains a grand challenge. Herein we report a series of 2-/2,6-aryl substituted boron-dipyrromethene dyes with wide-range and multi-fluorescence emissions across red and near infrared in their aggregation states. Experimental data of X-ray diffraction, UV–vis absorption, and room temperature fluorescence spectra have proved the multiple excitation and easy-adjustable emission features in aggregated boron-dipyrromethene dyes. Temperature-dependent and time-resolved fluorescence studies have indicated a successive energy transfer from high to step-wisely lower-located energy levels that correspond to different excitation states of aggregates. Consistent quantum chemical calculation results have proposed possible aggregation modes of boron-dipyrromethene dyes to further support the above-described scenario. Thus, this study greatly enriches the fundamental recognition of conventional boron-dipyrromethene dyes by illustrating the relationships between multiple emission behaviors and the aggregation states of boron-dipyrromethene molecules.

[1] Institute of Advanced Materials (IAM), Jiangsu National Synergetic Innovation Center for Advanced Materi/als (SICAM), Nanjing Tech University (NanjingTech), 30 South Puzhu Road, 211816 Nanjin, China. [2] Institute of Advanced Synthesis (IAS), School of Chemistry and Molecular Engineering, Jiangsu National Synergetic Innovation Center for Advanced Materials (SICAM), Nanjing Tech University (NanjingTech), 30 South Puzhu Road, 211816 Nanjing, China. [3] Institute of Advanced Materials (IAM), Jiangsu National Synergetic Innovation Center for Advanced Materials (SICAM), Nanjing University of Posts & Telecommunications, 9 Wenyuan Road, 210023 Nanjing, China. [4] State Key Laboratory of Coordination Chemistry, Collaborative Innovation Center of Advanced Microstructures, School of Chemistry and Chemical Engineering, Nanjing University, 210046 Nanjing, China. These authors contributed equally: Dan Tian, Fen Qi.  Correspondence and requests for materials should be addressed to Z.S. (email: zshen@nju.edu.cn) or to Z.L. (email: iamzpliu@njtech.edu.cn) or to L.H. (email: iamlhuang@njtech.edu.cn)

Due to the urgently increased demand in the areas of optoelectronic devices, energy conversion, luminescence sensors, bioimaging, disease diagnosis, and photodynamic therapy, luminescent materials with multiple emission capabilities across a wide wavelength range, e.g., from visible to near infra-red (NIR), as well as with tunable excitation/emission wavelengths have drawn rapidly increased attentions from researchers working in the subjects including but not limited to chemistry, physics, materials, and biology[1–4]. However, these materials are typically restrained within inorganic compounds such as metal-containing inorganic complexes particularly rare earth luminophors where the emissions usually come from the $f$–$f$ or $d$–$f$ electron transitions[5,6], or quantum dots whose size-dependent multiple luminescence emissions originate from the $d$-$d$ electron transitions. Unfortunately, the emission wavelengths of rare earth luminophors is very difficult to manipulate due to the shielding effect from exterior orbitals and the fixed energy levels of the rare earth ions while quantum dots suffer from both high toxicity of the heavy metals and environmentally sensitive emissions in the NIR region, which have prevented them from even wider applications. On the contrary, organic luminophors are more desired due to their competitive advantages such as arbitrarily tunable molecular structures and chemical compositions, easy for functionalization and scalable synthesis, and promising potentials to satisfy biomedical related and especially flexible electronics-oriented expectations.

Fluorescence emissions of organic luminophors are closely connected with their multiple electronic configurations and/or structures between the electronically ground ($S_0$) and excited ($S'$) states. Many strategies including but not limited to excited-state cis-trans isomerization[7,8], proton-transfer tautomerism[9–11], ring opening/closure isomerization[12], geometry planarization[13], and twisted intramolecular charge transfer[14], have been employed to create multiple energy levels between $S_0$ and $S'$ to generate multiple luminescence emissions. However, these methods are more effective in solution conditions and not working well in solid state because of the molecular aggregation-induced rigidified configuration, which does not favor multi-emissions with plausible efficiency. What is more, solid-state organic luminophors usually suffer from aggregation-caused emission quenching (ACQ) due to the large $\pi$–conjugation system formed[15]. Therefore, it remains an open challenge to obtain multi-emissions from organic luminophors in their solid state.

As a star molecule, BODIPY (4,4-difluoro-4-bora-3a,4a-diaza-s-indacene) dyes have been heavily investigated for bio-sensing and bioimaging thanks to their excellent luminescence performance in diluted solution state[16–18]. Since most BODIPYs suffer from ACQ, further manipulations such as decorating the classical BODIPY core (**BDP**) with bulky groups[19], asymmetrization of **BDP**[20,21], and aggregation-induced emission (AIE) are usually involved to achieve solid-state emissions and satisfy various requirements[22–24] while less attention has been paid on the fluorescence properties of BODIPYs in their aggregated state. Recent studies have shown that, contrast to their non-emissive H-aggregates[25], J-aggregates[26] of BODIPYs gave red-shifted emissions compared with their respective monomers in solution. For example, Johansson et al. evidenced the first formation of emissive J-dimers of BODIPYs in lipid vesicles[27] while Kim et al. and Chiara et al. demonstrated J-aggregation induced enhanced emission of BODIPYs[28–30]. Most importantly, J-aggregation of BODIPYs has provided a potential platform to achieve multi-emissions at longer wavelength, as reported by Jung[31], Harriman[32], and Yamamoto[33], respectively, where the green and red emissions from both monomer and J-dimer of BODIPYs in solid state have been obtained. However, in most cases only the emissions from J-dimer is seen and the multi-emissions is usually limited within the visible range while no report on other formats of BOIDPYs such as trimer or even larger J-aggregates in solid state[26–32].

Herein, we report that besides the yellowish green fluorescence emission of BOIDPY monomers in solution, microcrystalline powder of 2- and/or 6-aryl substituted BOIDPY (**BDP1** to **BDP6**) can generate significantly wide range of multi-emissions across red and NIR, particularly that they can be excited by light at multiple wavelengths (Fig. 1). Temperature-dependent and time-resolved fluorescence spectral results have suggested that J-aggregates with successively distributed and step-wisely lower-located energy levels are formed in the solid-state **BDP1** to

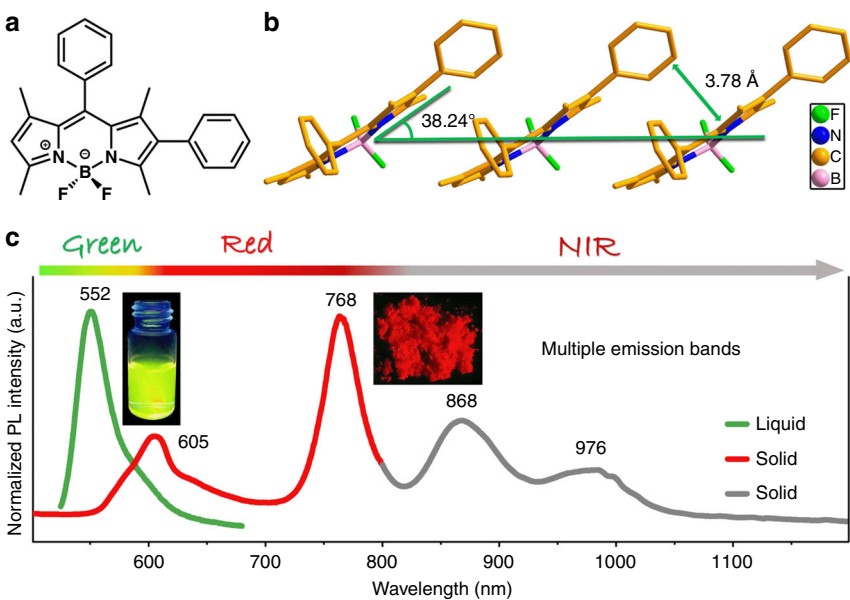

**Fig. 1** Structural details and fluorescent properties of BDP1. **a** Molecular structure of BDP1. **b** Molecular packing diagrams of BDP1 at room temperature extracted from single crystal XRD data. **c** Normalized photoluminescence (PL) spectra of BDP1 in solution (green curve) and microcrystalline powder state (red and gray curves)

**BDP6**, which facilitates Domino-like energy transfer at their respective excitation states. Moreover, the consistent quantum chemical calculation results on molecular packing modes have not only rationalized the above experimental phenomena, but also provided further guidance for our future working effort.

## Results

**Crystal structure**. Analysis of the X-ray diffraction (XRD) data (Supplementary Fig. 1) of single crystal **BDP1** indicates that the indacene plane is planar (Fig. 1a) and the dihedral angles between 2- and 8-phenyl substituents and indacene plane are 46.1° and 83.7°, respectively, suggesting that 2-phenyl ring has relatively better conjugating contribution to the indacene plane than that of the 8-phenyl ring (Supplementary Fig. 1). The boron atom in **BDP1** adopts a classical tetrahedral geometry within a six-membered ring (Fig. 1a), which shows a head-to-tail arrangement of boraindacene planes with adjacent 2-phenyl units interacted with a slipping angle ($\theta_J$) of 38.24° at room temperature (Fig. 1b). This is a typical feature for J-aggregates with long-range ordered structures[25,27–30] and the average vertical distance between parallel aromatic planes ($D_J$) is 3.78 Å, leading to commendable intramolecular electronic interactions (Fig. 1b). The distance between **BDP1** molecules was efficiently condensed at lowered temperatures due to the decreased molecular thermal vibration and $\theta_J$ and $D_J$ continuously decrease from 38.57°, 3.84 Å at 397 K to 38.04°, 3.71 Å at 187 K, until 37.87°, 3.68 Å at 87 K (Supplementary Fig. 2 and Supplementary Table 1).

**Photophysical property**. **BDP1** in tetrahydrofuran (THF) solution ($1 \times 10^{-5}$ mol L$^{-1}$) exhibits strong fluorescence emission at 552 nm (Fig. 1c). The 42 nm red-shift compared with that of **BDP** without the 2-phenyl ring at $\lambda_{em} = 510$ nm (Supplementary Fig. 3) suggests efficient conjugation between 2-phenyl ring and **BDP** frame in **BDP1** molecules[15,34], which agrees well with the small dihedral angles between 2-phenyl ring and indacene plane

extracted from the single crystal XRD data (Supplementary Fig. 1). More intriguingly, multi-emissions of **BDP1** in the powder of microcrystalline state at 605, 768, 868, and 976 nm (Fig. 1c) across red and NIR with decent fluorescence intensity ($\Phi_f = 0.1$) were observed under single-wavelength excitation (485 nm). The multi-absorption bands of **BDP1** (Supplementary Fig. 4) implies the existence of multi-excitation states corresponding to multiple energy levels, that is, the J-aggregates. For example, multiple excitation bands at 574, 680, and 749 nm are observed if 860 nm is fixed as the emission band (Supplementary Fig. 5), which not only proves the multi-wavelength excitation feature but also suggests the existence of multiple J-aggregates in **BDP1** with according excitation states. Indeed, upon excitations at 460, 600, 680, and 749 nm, multi-emission bands in the region of 550−1100, 700−1100, 700−1100, and 800−1100 nm can be routinely obtained (Supplementary Fig. 6), respectively. The identical emission bands appeared when excited by light at different wavelengths implies the existence of cascaded multiple excitation energy levels, which corresponds to the multiple J-aggregates formed in the powder of microcrystalline **BDP1**. Apparently, the continuous multi-emissions ranging from visible to NIR region are due to the successive energy transfer from high to step-wisely lower-located energy levels. Such sequential energy transfer behaves just like the falling row of dominoes and the characteristic Domino-like multiple fluorescence emission is principally different from those seen in rare earth luminophors or quantum dots that has fixed or size-dependent energy levels.

**BDP1** exhibits absorption maxima ($\lambda_{max}$) at 525 nm with a shoulder peak at 486 nm (Fig. 2a) and the strong maximum fluorescence emission ($F_{max}$) at 552 nm in THF solution with quantum yield close to unity ($\Phi_f = 0.92$). The small Stokes shift (27 nm) and high $\varepsilon$ ($8.09 \times 10^4$ M$^{-1}$ cm$^{-1}$) are typical characteristics of BODIPY dyes, which assures high value of the Förster radius ($50 \pm 2$ Å) for efficient energy transfer between each molecule (Homo-FRET)[26,35]. The position and intensity of both absorption and emission of **BDP1** in other solvents with different

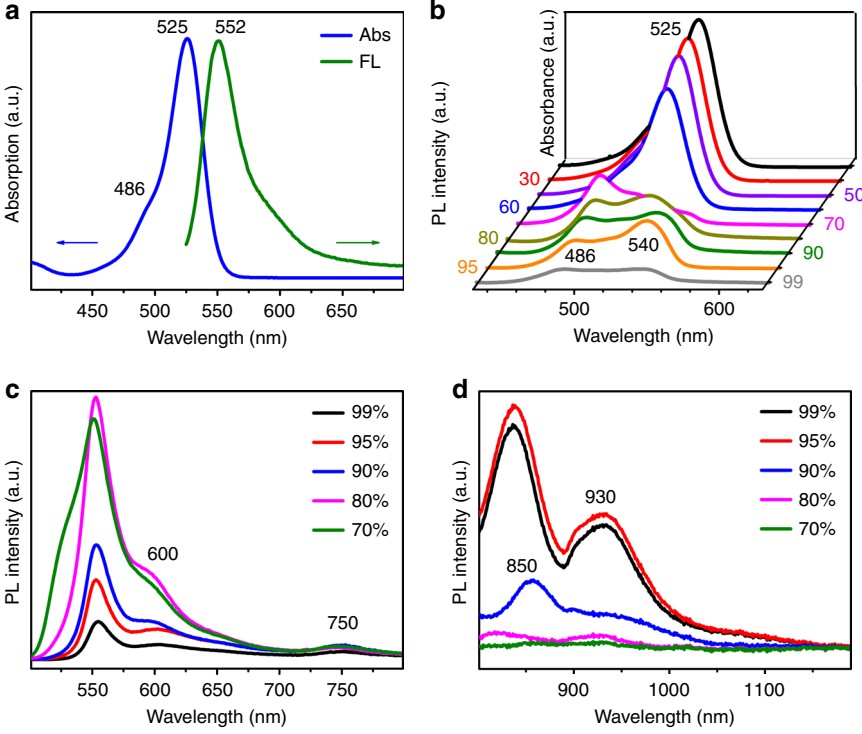

**Fig. 2** Photophysical properties of BDP1 in THF and THF/water mixtures. **a** Absorption and emission spectra of BDP1 in THF at $1 \times 10^{-5}$ mol L$^{-1}$. **b** Absorption and **c**, **d** emission spectra of BDP1 in THF/water mixtures with varied volumetric fractions of water ($f_w$)

polarities remain almost unchanged, suggesting the single molecular fluorescence behavior and no formation of J-aggregates in those solvents (Supplementary Fig. 7).

However, when water as a poor solvent was added into THF solution of **BDP1** at varied volumetric factions ($f_w$), the absorption maxima at 525 nm kept decreasing when $f_w$ increases from 0 to 60% (Fig. 2b). It then decreased tremendously and became broadened at $f_w$ of 70% where the 525 nm peak became undistinguishable. Moreover, the shoulder peak at 486 nm emerged and co-existed with a new peak at 540 nm until $f_w$ is at 99%. In the meanwhile, the fluorescence emission band at 552 nm became dramatically weakened when $f_w$ reaches 30, 50, and 60% (Supplementary Fig. 8) and then new peaks at 600, 750, 850, and 930 nm appeared at $f_w$ of 70–99% (Fig. 2c, d). This suggests that J-aggregates with multi-excitation states corresponding to the above emission peaks are starting to form at $f_w$ of 70%, which is reasonable because **BDP1** has poor solubility in water and more and more aggregates are formed at continuously increased value of $f_w$[22]. However, rather than ACQ, it is exciting to see that the J-aggregates formed at different values of $f_w$ are responsible for the above multiple and wide-range fluorescence emissions. The weakened and slightly blue-shifted emissions observed in Fig. 2c, d compared to those in microcrystalline powder (Fig. 1c) can be rationally attributed to the relatively poor packing order of **BDP1** molecules in THF/water.

To further disclose the multi-emission behavior of **BDP1** at different aggregation states, PMMA films doped with **BDP1** at varying concentrations have been prepared. Fig. 3a shows that

there is only one strong fluorescence emission with $F_{max}$ at 552 nm with the **BDP1** concentration ($c_{BDP1}$) of $2.5 \times 10^{-3}$ mol $L^{-1}$, which agrees very well with that of **BDP1** in THF solution (Fig. 2a) and suggests no aggregation of **BDP1** molecules at such low concentration. However, when $c_{BDP1}$ reaches $2.5 \times 10^{-2}$ mol $L^{-1}$, four new emission bands appeared at 580, 730, 817, and 915 nm. The emission band at 552 nm becomes continuously weakened along with the strengthened fluorescence intensity at 585, 735, 821, and 919 nm when $c_{BDP1}$ was further increased to $1.0 \times 10^{-2}$ mol $L^{-1}$. This indicates that **BDP1** molecules start to form J-aggregates at $c_{BDP1}$ of $2.5 \times 10^{-2}$ mol $L^{-1}$ and meanwhile free molecules are gradually depleted, which also explains why the fluorescence emission at 552 nm kept weakening and became almost disappeared at $2.5 \times 10^{-2}$ mol $L^{-1}$. Likewise, further increase of $c_{BDP1}$ to $3.0 \times 10^{-1}$ mol $L^{-1}$ causes even more red-shifted fluorescence emissions at 598, 742, 833, and 931 nm. The continuous red-shift of the above emissions at increased $c_{BDP1}$ suggests improved packing order of **BPD1** molecules in the aggregates that is more and more alike the case in the microcrystalline state shown in Fig. 1c, which is also exactly the same phenomenon observed at increased value of $f_w$ in Fig. 2c, d. Although with much lower intensity, it is worthy to point out that the emission band at 598 nm in PMMA film could be associated with that of 605 nm in the microcrystalline state in Fig. 1c, so do the bands at 742, 833, and 931 nm in Fig. 3a to those at 768, 868, and 976 nm in Fig. 1c, respectively. In other words, although the packing mode of **BDP1** molecules in PMMA film is not as perfect as that in the microcrystalline state, it is getting better at increased

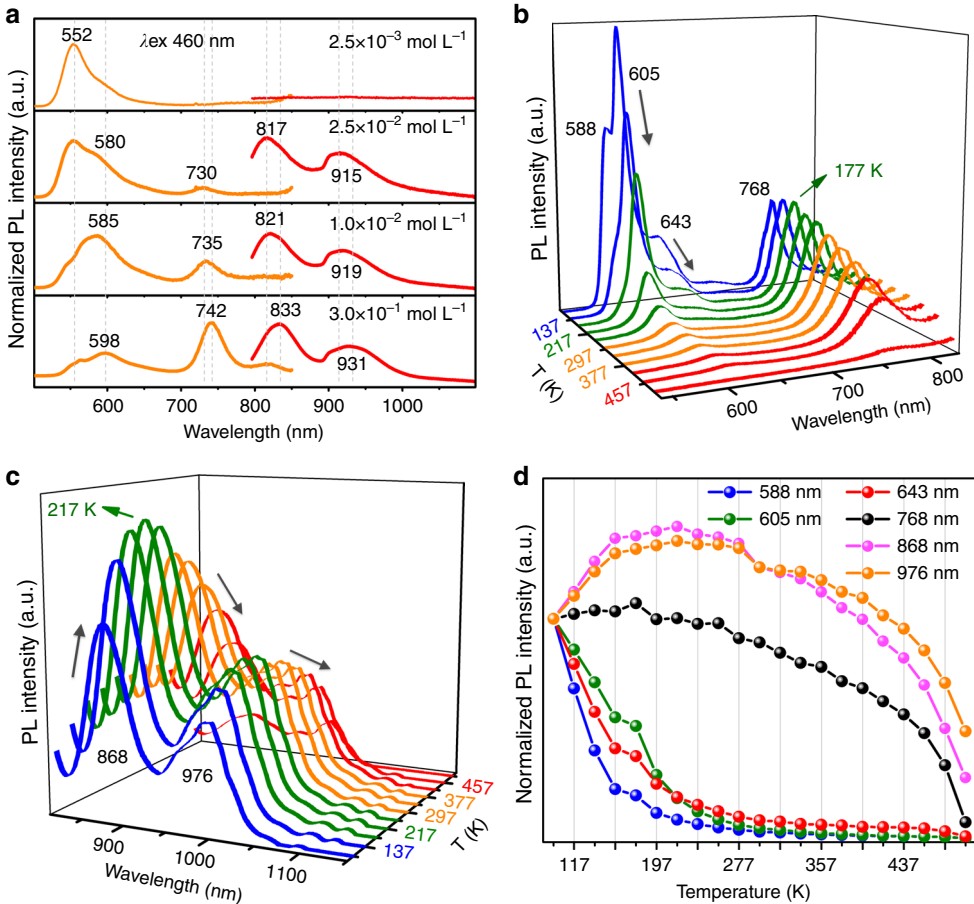

**Fig. 3** Fluorescent properties of aggregation-state BDP1. **a** Normalized fluorescence emission spectra of PMMA films prepared at different doping concentrations of BDP1. **b**, **c** Temperature-dependent emission spectra of microcrystalline powder state BDP1 collected from 97 to 497 K, excited at 480 nm. **d** Plots of normalized emission intensity changes of the multi-emissions at varying temperatures

doping concentrations as reflected by the gradually decreased blue-shift at increased $c_{BDP1}$ (Fig. 3a), comparing with the peaks seen in Fig. 1c.

Temperature-dependent fluorescence studies of microcrystalline **BDP1** have shown that the intensity of the major band at 605 nm keeps decreasing at increased temperature while the shoulder bands at 588 and 643 nm become unidentified at temperatures higher than 217 K (Fig. 3b). In the meanwhile, the intensity of 768 nm emission increases until reaching its maximum at 177 K and then starts decreasing at further elevated temperatures (Fig. 3b). Similarly, the increase prior to decrease of the fluorescence intensities are also seen in the emissions at 868 and 976 nm (Fig. 3c) at temperatures of 217 K, respectively. The temperature-dependent intensity variations of the according fluorescence emissions are plotted in Fig. 3d where the intensity of the emission bands at 588, 605, and 643 nm decreases monotonously while those at 768, 868, and 976 nm experience a slight increase at 97–177 K for 768 and 97–217 K for 868 and 976 nm emissions, respectively, followed by a similar monotonous decrease. As is well documented, the fluorescence emission usually decreases at elevated temperatures for organic luminophors due to the enlarged $\theta_J$ and $D_J$ as well as increased thermal vibration[34]. The observed intensity increase in the case of 768, 868, and 976 nm emissions clearly implies an exterior energy being pumped into these 3 excitation states where the energy states corresponding to the emissions at 588, 605, and 643 nm remain the only possibility.

Indeed, room temperature time-resolved fluorescence studies on the intensity and lifetime variations have confirmed the energy transfer from 605 to 768 nm emission. It is easy to see from Fig. 4a, b that the peak intensity at 605 nm reaches its maximum at 0.35 ns (Supplementary Fig. 9) and then decays rapidly with average lifetime of 0.92 ns (Supplementary Fig. 10 and

Supplementary Table 2), which is contributed by three lifetimes of 0.83 ns (55.6%), 2.89 ns (5.96%), and 1.88 ns (38.4%). Meanwhile, the 768 nm emission reaches its maximum at 1.40 ns with negative pre-exponential and a rise time with fitted value of 0.73 ns (Fig. 4b–d and Supplementary Table 2) was detected, which correlates closely with the short lifetime of 0.83 ns at 605 nm emission and straightforwardly confirms the energy transfer between these two excitation states.

Similarly, the increase prior to decrease of the emission intensity of 868 nm remains the same pace as the decrease of the 768 nm emission at low temperature was also observed although they could not be plotted in the same curve due to the instrument hardware limitation (Fig. 3b, c). Moreover, the perfect correlation of the intensity decrease of the 868 nm emission with the increase of the 976 nm emission (Fig. 3c) also indicates a similar energy transfer process. Meanwhile, time-resolved fluorescence emission at 976 nm behaves the same as that of 768 nm (Supplementary Fig. 11), suggesting a similar energy pumped-in event. What is more, the negative pre-exponential decay behavior was observed in 976 nm emission (Supplementary Table 2 and Supplementary Fig. 11), clearly suggesting that there exists a successive energy transfer process happened in turn from 605 to 768, 768 to 868, and finally 868 to 976 nm. Accordingly, there shall also exist multi-excitation states in the J-aggregates formed in microcrystalline **BDP1** at different aggregation states, which facilitates the Domino-like multi-emissions.

All the fluorescence spectroscopic data in THF/Water mixture, PMMA doped film, and microcrystalline state have consistently proved that the multi-excitation and multi-emission phenomena are intrinsic fluorescence behaviors of **BDP1** where the emission wavelength, intensity, and decay time are closely relevant to the aggregation states of **BDP1** molecules. Moreover, the vibrational shoulder bands of 588 and 643 nm centered at 605 nm observed

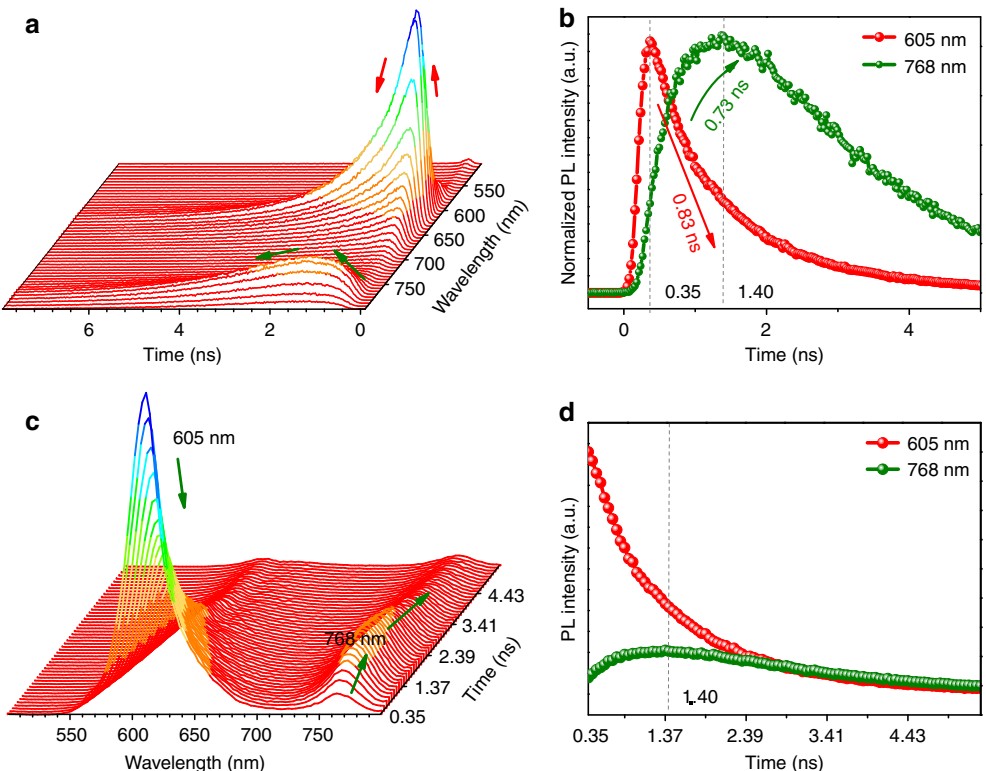

**Fig. 4** Time-resolved fluorescence properties of microcrystalline state BDP1. **a** Time-resolved fluorescence decay curves of BDP1 at emission wavelength ($\lambda_{em}$) of 500–800 nm. **b** Fluorescence decay curves of BDP1 at emission wavelengths of 605 and 768 nm. **c** Time-resolved emission spectra of BDP1 at the detection time from 0.35 to 5.45 ns **d** Time-resolved emission intensity variation of BDP1 at 605 and 768 nm. All samples measured were in microcrystalline powder state

at 97 to 217 K is indicative of the existence of **BDP1** monomers, and the 53 nm redshift compared to the monomer ($\lambda_{em} = 552$ nm) in THF solution is due to the improved $\pi$-conjugation and strong intermolecular interactions in the microcrystalline powder state[14,34]. No shoulder bands, different fluorescence decay times, and sequential energy transfer strongly suggest that the formation of different aggregates such as dimer, trimer, etc. are responsible for the emissions observed at 768, 868, and 976 nm, respectively.

**Theoretical calculation**. Quantum chemical calculations were further performed to illustrate the Domino-like energy transfer process existed in the aggregation induced J-aggregates from different point of view. Starting from the crystalline structure of **BDP1** (Supplementary Fig. 12), several multimers (three dimers and three trimers) were constructed from nearby **BDP1** monomers, and their excitation state parameters including excitation energy, oscillator strengthen ($f$), and natural transition orbitals (NTOs) of the lowest singlet state ($S_1$) were calculated using the TDDFT/M06-2X/6-31G(d) method (Supplementary Fig. 13 and Supplementary Table 3).

The computational results have indicated that: (1) the $S_1$ of **BDP1** monomer ($f = 0.647$) is stemmed from the local transition of BODIPY moiety, that is, the NTOs in Supplementary Fig. 13. The calculated excitation energy (2.960 eV) overestimates the experimental value (2.363 eV, 525 nm in Fig. 2a) with 0.597 eV, which is a systematic shift (greater than 0.3 eV) of TD-DFT results compared to experimental ones for BODIPY dyes;[36] (2) both holes and particles in dimer-1 and trimer-1 are equally delocalized over **BDP1** moieties between **BDP1** monomers (Fig. 5a, b), forming a strong state interaction with small redshift (up to 0.048 eV for trimer-1) and increased oscillator strengthens ($f = 1.263$ for dimer-1 and $f = 0.654$ for trimer-1). These results are consistent with the variation trend seen in experiment, namely, the redshift (0.065 eV) and enhanced absorption band at 540 nm (2.298 eV) in THF-water mixtures (greater than 70%) compared with the monomeric absorption at 525 nm (2.363 eV); (3) the hole-particle distribution of other calculated multimers (dimer-2, dimer-3, trimer-2, and trimer-3) shows similar transition behavior as those of monomers (Supplementary Fig. 13), which will not contribute to the emissions at longer wavelengths seen in Figs. 1c and 3a.

Combining the theoretical and experimental results, the emission band at 605 nm can be assigned as monomeric emission, while the band at 768, 868, and 976 nm may originate from multimers such as dimer-1, trimer-1, etc., respectively. Consequently, possible fluorescence mechanism with the according energy transfer between the multi-excitation states of **BDP1** in different aggregation states is proposed in Fig. 5c.

**Scope of multiple emissions of BDP1 analogues**. More importantly, further investigations have proved that the analogues of **BDP1**, that is, **BDP2** to **BDP6**, also possess similar multi-excitation (Supplementary Fig. 14) and multi-emission (Fig. 6) capabilities in their respective microcrystalline powder states. Besides the typical emission band in the region of 600 to 670 nm from corresponding monomers, **BDP2** to **BDP6** show characteristic NIR emissions of their aggregates at 753, 899, and 980 nm for **BDP2**, 764, 888, and 968 nm for **BDP3**, 717 and 907 nm for **BDP4**, 704 and 905 nm for **BDP5**, and 749, 833, and 900 nm for **BDP6** (Fig. 6), respectively. These results directly imply that such multi-emission behaviors of **BDP1** to **BDP6** shall be a general case existed not only in its analogues but also in all BODIPY derivatives when substituted with proper functional groups.

## Discussion

In conclusion, we have systematically investigated the multi-excitation and multi-emission phenomena observed in the solid-state **BDP1** to **BDP6**. The proposed aggregation-associated Domino-like energy transfer mechanism is well-supported by temperature-dependent and time-resolved fluorescence studies as well as the quantum chemical calculation results. The interesting emission features such as tunable and wide range multi-emissions ranging from green to NIR will impart **BDPs** greater potential than their analogues in the areas of flexible optoelectronics, frequency-controlled lighting, anticounterfeiting, optical communications, and bioimaging and disease diagnosis. This initial discovery will encourage further exploration on emission behaviors and mechanisms of not only BODIPYs but also other conventional organic dyes in their proper aggregation states. We anticipate that this study will pave an avenue for further

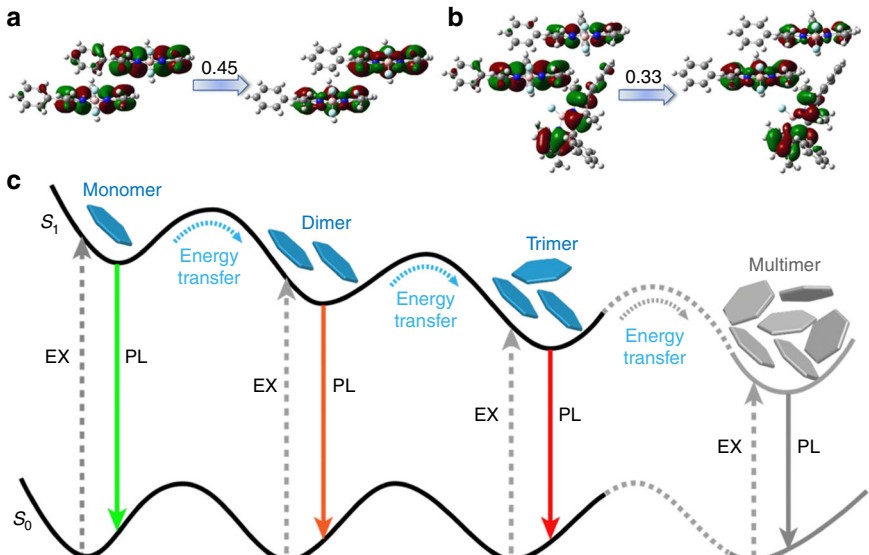

**Fig. 5** Theoretical calculations for mechanistic investigation. Calculated NTOs of BDP1 **a** dimer-1 and **b** trimer-1. **c** Schematic illustration of the proposed Domino-like energy transfer and subsequent multiple emissions in solid state BDP1

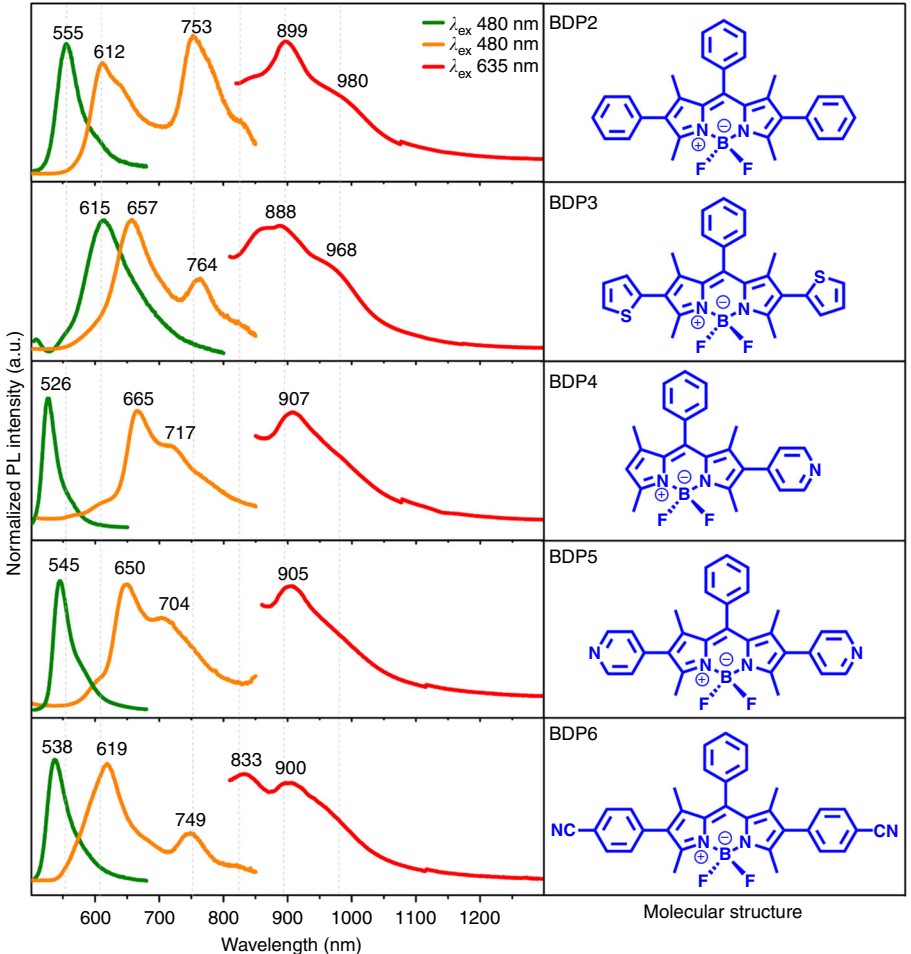

**Fig. 6** Photophysical properties of microcrystalline state BDP2 to BDP6. Normalized fluorescence emissions (left column) of **BDP2** to **BDP6** in solution (green lines) and microcrystalline powder state (orange and red lines) with corresponding molecular structures in the right column

manipulation of multiple energy levels of solid-state organic luminophors toward a greatly widened scope of applications.

## Methods

**Reactions and purifications**. All the air-sensitive synthesis were operated in the argon environment by using Schlenk vacuum line techniques. Pre-dried solvents were used in the organometallic reactions. Commercially purchased chemicals were used as received. Thin column chromatography was carried out using 200 mesh silica gel (Qingdao Haiyang Chemical Co., Ltd). **BDP2**[37], **BDP3**[36], and **BDP5**[38] were synthesized according to the reported literatures. [1]H NMR (Supplementary Figs. 15, 17 and 19) and [13]C NMR (Supplementary Figs. 16, 18 and 20) data of **BDP1**, **BDP4**, and **BDP6** were included in the Supplementary Methods.

Synthesis of **BDP1**: 2-iodo-1,3,5,7-tetramethyl-8-phenyl–4,4-difluoroboradiazaindacene (200 mg, 0.44 mmol), phenylboronic acid (81 mg, 0.66 mmol), and tetrakis(triphenylphosphine) palladium(0) (23 mg, 0.02 mmol) were added to a round-bottomed flask connected with a reflux condenser and 2 mL of toluene was then added. The reaction mixture was heated at 90 °C for 1 d followed by the addition of 2 mL of 1 mol L$^{-1}$ sodium carbonate aqueous solution. After cooling down to room temperature, the as-obtained crude mixture was extracted with dichloromethane (DCM), dried with anhydrous magnesium sulfate, and then purified through the silica gel column chromatography (hexane: DCM, 2: 1, v: v, $R_f$ = 0.3). Yield: 158 mg (90%).

Synthesis of **BDP4**: 2-iodo-1,3,5,7-tetramethyl-8-phenyl-4,4-difluoroboradiazaindacene (200 mg, 0.44 mmol), 4-(4,4,5,5-tetramethyl-1,3,2-dioxaborolan-2-yl)pyridine (145 mg, 0.66 mmol), and tetrakis(triphenylphosphine) palladium(0) (23 mg, 0.02 mmol) were added to a round-bottomed flask connected with a reflux condenser and 2 mL of toluene was then added. After adding, The reaction mixture was heated at 90 °C for 1 d followed by the addition of 2 mL of aqueous 1 mol L$^{-1}$ sodium carbonate aqueous solution. After cooled down to room temperature, the as-obtained crude mixture was extracted with DCM, dried with anhydrous magnesium sulfate, and then purified through the silica gel column chromatography (ethyl acetate: DCM, 1: 10, v: v, $R_f$ = 0.4). Yield: 162 mg (92%).

Synthesis of **BDP6**: 2,6-diiodo-1,3,5,7-tetramethyl-8-phenyl-4,4-difluoroboradiazaindacene (200 mg, 0.35 mmol), (4-cyanophenyl)boronic acid (123 mg, 0.84 mmol), and tetrakis(triphenylphosphine) palladium(0) (51 mg, 0.04 mmol) were added to a round-bottomed flask connected with a reflux condenser and 2 mL of toluene was then added. The reaction mixture was heated at 90 °C for 1 d followed by the addition of 2 mL of 1 mol L$^{-1}$ sodium carbonate aqueous solution. After cooled down to room temperature, the as-obtained crude mixture was extracted with DCM, dried with anhydrous magnesium sulfate, and then purified through the silica gel column chromatography (DCM, $R_f$ = 0.5). Yield: 150 mg (65%).

**X-ray crystallographic analysis**. Red and rectangular single crystal of **BDP1** was crystallized from a solution of DCM. Intensity data of **BDP1** (87, 187, 297, 397 K) were collected on a Bruker SMART APEX-II CCD X-ray diffractometer with Mo-Kα radiation ($\lambda$ = 0.71073 Å). The structure was interpreted and refined by SHELX-2014. The crystallographic parameters are summarized in Supplementary Table 4.

**Absorption and fluorescence spectroscopic studies of BDP1 in solution**. UV-vis absorption spectra in solution ($1 \times 10^{-5}$ mol L$^{-1}$) in a 1 cm quarts cuvette were collected on a Shimadzu UV-1750 spectrometer at resolution of 1.0 nm. Fluorescence spectra in solution were recorded on a Horiba Jobin Yvon spectrometer (Nanolog FL3-2iHR). The luminescence quantum yield in solution was measured using rhodamine 6G (under excitation of 488 nm, $\Phi_f$ = 0.88 in ethanol) as reference. The quantum yield $\Phi$ is calculated using the equation: $\Phi_u = [(A_s F_u n^2)/(A_u F_s n_0^2)]\Phi_s$, where $A_s$ and $A_u$ are the absorbance of the reference (or standard) and sample (or unknown) solutions at their respective excitation wavelengths, $F_s$ and $F_u$ are the corresponding integrated fluorescence intensity, and $n$ and $n_0$ are the refractive indexes of the solvents of the sample and the reference, respectively[39].

**Absorption and fluorescence spectroscopic studies of solid-state BDP1**. Microcrystalline powder samples of **BDP1** were used for the determination. The solid-state UV–vis absorption spectra ranging from 200 to 1200 nm were measured

on a Lambda 950 UV–vis spectrophotometer (PerkinElmer, U.S.A.), and using $BaSO_4$ as the reference sample. 0.5 g microcrystalline powder samples of **BDP1** was pressed on $BaSO_4$ by the tableting method. The temperature-dependent and steady-state fluorescence spectra were collected on a Horiba Jobin Yvon spectrometer (Nanolog FL3-2iHR) and corrected against photomultiplier and lamp intensity. For the NIR spectra, light filter (780 nm) was used to remove the scattering light from primary frequency and frequency multiplication. The absolute quantum yields ($\Phi_f$) was obtained on a Horiba Jobin Yvon spectrometer (Nanolog FL3-2iHR). The fluorescence lifetimes were determined on a Horiba Jobin Yvon IBH Inc spectrometer (Delta flex).

**PMMA films preparation**. 100 mg poly(methyl methacrylate) (PMMA, 120000 average molecular weight, density $d_{PMMA} = 1.188$ g cm$^{-1}$) was dissolved in DCM ($V_{DCM} = 10$ mL). DCM solutions of **BDP1** with different concentrations were then mixed with PMMA solution ($V_{PMMA} = 1$ mL) for 30 min under ultrasonic stirring. The standard spin-coating technique was used to prepare the films of PMMA containing **BDP1**. Firstly, glass microscope slides (cut to 25 mm × 10 mm × 1 mm) were washed with ethanol, wipe-cleaned, and placed on a spin coater; Secondly, 40 μL of the dye/PMMA mixture in DCM was added using a pipette on the glass slide, and the film was formed when spin-coating at 3000 rpm for 60 s.

**Fluorescence spectroscopic studies of BDP1 in PMMA films**. Fluorescence spectra in PMMA films were collected on a Horiba Jobin Yvon spectrometer (Nanolog FL3-2iHR).

**Time-resolved fluorescence spectroscopy**. All the time-resolved fluorescence spectra (TRFS) were obtained on UltraFast lifetime Spectrofluorometer (Delta flex, Horiba Jobinyvon IBH Inc). The excitation source was provided by a DeltaDio-de$^{TM}$ (Horiba) pulse laser diode with wavelength of 485 nm. TRFS were obtained by uniformly incrementing of the wavelength and measuring the time-resolved decay for equal periods of time. Fluorescence decays were then analyzed using DAS6 software by fitting common lifetimes to all of the individual decay curves.

**Theoretical modeling**. The computational models were built from the crystal structure shown in Supplementary Fig. 12. The quantum/molecular mechanics (QM/MM) theory with two-layer ONIOM method was implemented to deal with the molecular electronic structures in the crystal where the central molecule is chosen as the active QM part and defined as the high layer while the surrounding ones are chosen as the MM part and set as the low layer. The universal force field is applied for the MM part and that of the molecules of MMare frozen during the QM/MM geometry optimization. The electronic embedding is used in QM/MM calculations by combining the partial charges of the MM region with the QM Hamiltonian. The M06-2X functional has been proposed to rationally describe the BODIPY dyes[40] and deal with the weak intermolecular interactions (van der Waals and π–π coupling)[41]. Several test configurations (three dimers and three trimers) were extracted from the **BDP1** crystal structure and their natural transition orbitals (NTO) and vertical excitation energies were then calculated at TDDFT/M06-2X/6-31G(d) level. Moreover, the lowest singlet states ($S_1$) of the possible dimer and trimer configurations were also optimized at the same level. All of the calculations were performed using the Gaussian 09 program[42].

**Data availability**. The data supporting the findings of this study are available from the authors on reasonable request, see author contributions for specific data sets. The X-ray crystallographic coordinates for the structures reported in this article have been deposited at the Cambridge Crystallographic Data Centre (CCDC) under deposition numbers CCD 1540363, 1540364, 1837300, and 1837301. These data can be obtained free of charge from The Cambridge Crystallographic Data Centre via www.ccdc.cam.ac.uk/data_request/cif.

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

## Acknowledgements

This work is supported by the National Key R&D Program of China (2017YFA0207201), the National Nature Science Foundation of China (Grant Nos. 21601086, 2151085, 21371095, and 21503118), the Natural Science Foundation of Jiangsu Province (BK20160994 and BL2014075), and Key University Science Research Project of Jiangsu Province (No. 17KJA150004). We thank Dr. Yan Guan (Analytical Instrumentation Center, Peking University) for time-resolved characterization and helpful discussion and Dr. Di Li (Institute for Energy Research, Jiangsu University) for figure construction using 3D Max software.

## Author contributions

D.T., F.Q., X.W., Z.S., Z.L., L.H. and W.H. conceived the experiments. Z.L., L.H., D.T. and Z.S. prepared the manuscript. D.T., F.Q. and Y.P. performed the experiments. H.M. performed the quantum chemical calculations. All authors discussed the results and commented on the manuscript.

## Additional information

**Competing interests:** The authors declare no competing interests.

