## [Peer Review File · Nature Communications]

Reviewers' Comments:

Reviewer #1:

Remarks to the Author:

In this paper, the authors describe that newly synthesized 2-/2,6-aryl substituted BODIPY dyes show multiple excited states by light irradiation in the solid state through their multiple J-aggregations. Surprisingly, the compound show green color emission in solution but, the wavelength of the solid-state-emission band of the same molecule reaches the infrared region. As the mechanism of the emission, the authors assume energy transfer between aggregation states from those with higher energy one to lower energy one. They also show experiments to support the mechanism for luminescence from the aggregation states. The authors prepared a sample where the molecules gradually aggregate by adding water to a solution sample of the molecule (BDP 1) in THF as a good solvent. Furthermore, experiments are conducted to compare optical properties by dissolving BDP 1 with different concentrations to PMMA. By performing time-resolved fluorescence measurement of each sample, the authors showed the possibility of the energy transfer between the aggregates. All experiments correspond to the assumption of the authors. To the best of our knowledge, BODIPY compounds that show IR emission by forming aggregation has been not known. However, there are several concerns as shown below.

1. The authors stated that there are multiple J-aggregates in the solid state, but are not mentioned in their structure details. Illustrations for the energy transfers between aggregations based on the single crystal structure analysis should be provided. The schematic structure of "multiple aggregations" should be illustrated.
2. The authors call this phenomenon "a typical Domino-like fluorescence emission behavior," but neither definition nor proper citation is shown. They should provide more detailed discussion about this with previous studies.
3. The meaning of the sentence of " This is caused by the continuous energy transfer from high to step-wisely lower-located energy levels of multiple J-aggregates" is unclear. In addition to improving this sentence, explain the phenomena using model figures for reader's understandings. S12 is too simple to understand. As mentioned above, illustrations including the molecular structures corresponding to S12 is necessary for the good understanding of the reader.
4. "It is worthy to point out that determined by the packing mode of BDP1 molecules at varied doping BDP1 concentrations". I cannot understand the meaning of this sentence. What does "Doping" mean?
5. Experimental details should be added to the absorption spectrum of the crystals shown in Figure S4. In general, it is not easy to take absorption spectra of crystals.

Reviewer #2:

Remarks to the Author:

This manuscript demonstrates multiple-absorption and emission characteristics of BODIPY dyes BDP 1-6. Most significant finding claimed in this work is the 'different excitation states of J-aggregates' which embodies the cascade energy transfer between them. Unfortunately, I find no detailed description of these states in this work. It appears that authors consider multiple J-aggregates with different coherency, which is hard to imagine in a tightly packed single crystalline structure. I think this manuscript is still incomplete in being unable to clarify those states unambiguously. Moreover, the suppressed luminescence quenching in J-stacked fluorophore and the level of PLQY ~0.1 are no more surprise in the literature. Therefore, I am unable to recommend publication of this work in Nature Communication.

Reviewer #3:

Remarks to the Author:

The authors report the intriguing multiple emissions across the red-edge of the visible (even reaching the NIR) of simple BODIPYs in the solid state. To unambiguously check such performance, they measure the photophysical properties not only in solid state, but also in solution (adding increasing amount of water) and in PMMA films, and in all cases they record similar multi-emissions owing to long-range J-aggregates and ongoing domino-like energy transfer. The presence of such kind of aggregation is well supported by absorption, excitation and fluorescence spectra (even considering temperature effect), as well as by time-resolved fluorescent measurements and X-ray diffraction data.

Actually, the solid-state emission is an active topic of research in BODIPY dyes, as reflected by the authors in the references. However, for the best of my knowledge is the first time that such multiple emissions reaching the NIR and with a fluorescence efficiency around 10%. These results are really astonishing and attractive to develop smart dyes emitting in the solid state with a fluorescence performance similar or even better than inorganic materials. The photophysical characterization is exhaustive and rigorous (the figures are very illustrative and the ESI is well documented) and the conclusions are well supported and are fully consistent.

Therefore, I find that the manuscript meet the required criteria of impact and novelty to deserve publication in Nature Communications.

Response to Reviewers' comments:

All changes mentioned in the response letter were accordingly red-marked in the revised manuscript and Supporting Information (SI).

Reviewer #1

In this paper, the authors describe that newly synthesized 2-/2,6-aryl substituted BODIPY dyes show multiple excited states by light irradiation in the solid state through their multiple J-aggregations. Surprisingly, the compound show green color emission in solution but, the wavelength of the solid-state-emission band of the same molecule reaches the infrared region. As the mechanism of the emission, the authors assume energy transfer between aggregation states from those with higher energy one to lower energy one. They also show experiments to support the mechanism for luminescence from the aggregation states. The authors prepared a sample where the molecules gradually aggregate by adding water to a solution sample of the molecule (BDP 1) in THF as a good solvent. Furthermore, experiments are conducted to compare optical properties by dissolving BDP 1 with different concentrations to PMMA. By performing time-resolved fluorescence measurement of each sample, the authors showed the possibility of the energy transfer between the aggregates. All experiments correspond to the assumption of the authors. To the best of our knowledge, BODIPY compounds that show IR emission by forming aggregation has been not known. However, there are several concerns as shown below.

Response: Thanks a lot for the positive comments and we really appreciate that.

1. The authors stated that there are multiple J-aggregates in the solid state, but are not mentioned in their structure details. Illustrations for the energy transfers between aggregations based on the single crystal structure analysis should be provided. The schematic structure of “multiple aggregations” should be illustrated.

Response: Thanks a lot for the constructive comments. We have added illustrations for the energy transfer between aggregations based on the single crystal structure analysis (Pages 7 and 10 in manuscript). Moreover, we have carried out theoretical calculations and come up with the schematic structures of the multiple aggregation states (please see Figures 5 in manuscript, Figure S12 and Table S3 in SI), which are consistent results to further support our explanations.

2. The authors call this phenomenon "a typical Domino-like fluorescence emission behavior," but neither definition nor proper citation is shown. They should provide more detailed discussion about this with previous studies.

Response: Thanks a lot for the critical comment. We have added description of "a typical Domino-like fluorescence emission behavior," in the revised manuscript (Page 7) and red-marked.

In addition, we have cited previous studies in our manuscript about such continuous energy transfer existed in the J-aggregates of BODIPY dyes. For example, energy transfer from the monomer of BODIPY to J-dimer reported by Méallet-Renault et al. (*J. Phys. Chem. C* **117**, 5373–5385 (2013); Ref. 31 in manuscript) and cavity-mediated long distance and multistep energy transfer of polystyrene microspheres doped with BODIPY reported by Yamamoto et al. (*ACS Nano* **10**, 7058–7063 (2016); Ref. 32 in manuscript). However, the continuous and Domino-like multistep energy transfer existed in various J-aggregates of solid state **BDP**, especially the multi-emission in across red and NIR region has not been reported before.

3. The meaning of the sentence of " This is caused by the continuous energy transfer from high to step-wisely lower-located energy levels of multiple J-aggregates" is unclear. In addition to improving this sentence, explain the phenomena using model figures for reader's understandings. S12 is too simple to understand. As mentioned above, illustrations including the molecular structures corresponding to S12 is necessary for the good understanding of the reader.

Response: Thanks a lot for the comments. We have re-described the sentence of " This is caused by the continuous energy transfer from high to step-wisely lower-located energy levels of multiple J-aggregates" (please see Page 7 in manuscript) and further explained the phenomena with more details using a proposed molecular aggregation structure obtained via theoretical modeling (please see Figures 5 in manuscript, Figure S12 and Table S3 in SI) in the revised manuscript and red-marked.

Yes, we agree that original S12 is too simple and we have replaced it with Figure 5 (and Figure S12, Table S3, SI) in the revised manuscript, which could provide more information on both the possible molecular packing mode and the multiple energy transfer details. Illustrations including the molecular structures of **BDP** at varying aggregation states corresponding to original S12 has also been provided in Figure 5 (Page 16) in the revised manuscript.

4. “It is worthy to point out that determined by the packing mode of BDP1 molecules at varied doping BDP1 concentrations”. I cannot understand the meaning of this sentence. What does “Doping” mean?

Response: We are sorry for the confusing description. “*Doping*” here means adding (certain amount of) **BDP1** molecules into the PMMA solution. “*Doping*” is widely employed in such studies. For example, **Lennart B.-Å. Johansson** has “achieved a locally high concentration of BODIPY groups by *doping* lipid phases (micelles, vesicles) with BODIPY-labeled lipids” in *J. Am. Chem. Soc.* **124**, 196-204 (2002), **Qingdong Zheng** reported “BODIPY-*doped* silica nanoparticles with reduced dye leakage and enhanced singlet oxygen generation” in *Sci. Rep.* **5**, 12602 (2015), and **Mark E. Thompson** studied the photoluminescence behavior of Pt(^{BDP}TPBP) *doped* into a poly(methyl methacrylate) (PMMA) matrix in *J. Am. Chem. Soc.* **133**, 88-96 (2011).

5. Experimental details should be added to the absorption spectrum of the crystals shown in Figure S4. In general, it is not easy to take absorption spectra of crystals.

Response: We have added experimental details of the UV-vis absorption spectrum of the crystals shown in Figure S4 and red-marked in the Supporting Information. Indeed, it is not easy to take absorption spectra of crystals. However, we have managed to do so and we are happy to obtain the valuable data to support our explanation.

Reviewer #2

This manuscript demonstrates multiple-absorption and emission characteristics of BODIPY dyes BDP 1-6. Most significant finding claimed in this work is the ‘different excitation states of J-aggregates’ which embodies the cascade energy transfer between them. Unfortunately, I find no detailed description of these states in this work. It appears that authors consider multiple J-aggregates with different coherency, which is hard to imagine in a tightly packed single crystalline structure. I think this manuscript is still incomplete in being unable to clarify those states unambiguously. Moreover, the suppressed luminescence quenching in J-stacked fluorophore and the level of PLQY ~0.1 are no more surprise in the literature. Therefore, I am unable to recommend publication of this work in Nature Communication.

Response: Thanks a lot for the critical comments. We are sorry for not being able to provide sufficient description of the “different excitation states of J-aggregates” and the according “cascade energy transfer between them” in the submitted version of our manuscript.

In the revised manuscript, we have added more descriptions on both the different excitation states of J-aggregates and the cascade energy transfer between the J-aggregates formed in solid state **BDP1** (please see Pages 7 and 10). Moreover, to gain more information on the packing mode of **BDP** molecules at varying aggregation states, we have carried out quantum chemical calculation experiment and the results obtained are consistent with the experimental ones. The molecular packing mode drawn from the calculation well-supports our experimental results. Detailed description of the molecular packing mode of **BDP1** molecules in the solid state as well as the energy transfer between the multi-excitation states can be found in Pages 14-16 and Figure 5 in the revised manuscript (please also see Figure S12 and Table S3, SI).

In terms of the PLQY, we want to say that ~0.1 is relatively high compared to previous studies. For example, J-aggregation of BODIPY dyes reported by Chiara et al. (*Adv. Funct. Mater.* **26**, 2756–2769 (2016); Ref. 29 in manuscript) and Yamamoto et al (*ACS Nano* **10**, 7058–7063 (2016); Ref. 32 in manuscript) still suffer from strongly quenched emission (PLQY<0.1) compared with their monomer in solution (PLQY>0.8). Even with the assistance of Aggregation-Induced Emission (AIE), the PLQY for J-aggregates of a meso-trifluoromethyl BODIPY reported by Kim et al is only 0.06 (*Chem. Sci.* **5**, 751–755 (2014); Ref. 27 in manuscript).

Moreover, according to the comments from **Reviewer #3** that “*for the best of my knowledge is the first time that such multiple emissions reaching the NIR and with a fluorescence efficiency around 10%.*” and meanwhile **Reviewer #1** has also stated “*To the best of our knowledge, BODIPY compounds that show IR emission by forming aggregation has been not known.*”, the most important finding of our manuscript is the first report of the continuous and Domino-like multistep energy transfer existed in various J-aggregates of solid state **BDP**, especially the multi-emissions across red and NIR region has not been reported before. Although at its preliminary stage, we believe that it represents an important advance in the luminescence study of solid state **BDPs**. We also believe that the PLQY can be further improved by adjusting the molecular structure, the packing mode of the molecules, the aggregation states of **BDP** molecules, or other effective means when later on we know more about this new phenomena. We firmly anticipate that this achievement will stimulate new areas of investigation and illuminate many aspects of research on **BDP** dyes relevant to lighting, disease diagnosis, and energy

conversion and so on. Certainly, the above-described directions also represent our future working efforts. We hope the reviewer can concur.

Reviewer #3

The authors report the intriguing multiple emissions across the red-edge of the visible (even reaching the NIR) of simple BODIPYs in the solid state. To unambiguously check such performance, they measure the photophysical properties not only in solid state, but also in solution (adding increasing amount of water) and in PMMA films, and in all cases they record similar multi-emissions owing to long-range J-aggregates and ongoing domino-like energy transfer. The presence of such kind of aggregation is well supported by absorption, excitation and fluorescence spectra (even considering temperature effect), as well as by time-resolved fluorescent measurements and X-ray diffraction data.

Actually, the solid-state emission is an active topic of research in BODIPY dyes, as reflected by the authors in the references. However, for the best of my knowledge is the first time that such multiple emissions reaching the NIR and with a fluorescence efficiency around 10%. These results are really astonishing and attractive to develop smart dyes emitting in the solid state with a fluorescence performance similar or even better than inorganic materials. The photophysical characterization is exhaustive and rigorous (the figures are very illustrative and the ESI is well documented) and the conclusions are well supported and are fully consistent.

Therefore, I find that the manuscript meet the required criteria of impact and novelty to deserve publication in Nature Communications.

Response: Thank you very much for the encouraging comments and we really appreciate that.

Reviewers' Comments:

Reviewer #1:

Remarks to the Author:

I found all concerns I raised have been addressed.

Reviewer #2:

Remarks to the Author:

In this revised manuscript, authors proposed a very specific emitting species (J-dimer, trimer, tetramer etc....) to explain the multiple emissions. While the assumption of such different coherence length (aggregation number) partly explains the observed multiple emission and relevant cascade energy transfer between them as experimentally demonstrated in this work, why different aggregation states are simultaneously existing in a structurally uniform single crystal is very puzzling. Although the authors made a NTO calculation based on the J-dimer, J-trimer, etc to show different S1 energy level of them to interpret the experimental result, arbitrarily extracting the isolated dimer (or trimer etc) geometry from the single crystal structure is not justified as to represent the whole crystalline bulk. If the authors want to unambiguously claim that the emitting species are the J-aggregates with different aggregation numbers, the proportion and nature of each species should be uncovered and also why and how they are formed in such a structurally regular single crystal should be rationalized. With such puzzling task unanswered, this work remains incomplete and unsupported. Therefore, I am unable to recommend publication of this work in Nat. Commun.

Response to Reviewers' comments:

All changes mentioned in the response letter were accordingly red-marked in the revised manuscript and Supporting Information (SI).

Reviewer #1

I found all concerns I raised have been addressed.

Response: Thanks a lot for the positive comments and we really appreciate that.

Reviewer #2

1. Why different aggregation states are simultaneously existing in a structurally uniform single crystal is very puzzling.

Response: We think it is obvious that Reviewer 2 has made a mistake in understanding the **quantum phenomenon** of the *multiple emissions generated from the transient molecular configurations at excited state* described in our manuscript from the **classical physics** point of view of the *frozen molecular structure* of **BDP1** molecules in the solid state. This is coincidentally consistent with Prof. Aprahamian's statement on the photophysical mechanism of solid-state organic luminescence emissions in his recent publication (*Nat. Chem.* **9**, 83-87 (2017)) that: "*However, understanding of the quantum-mechanical origin of a relationship between the structural phenomenon (the intramolecular motion) and the photophysical phenomenon (the AIE) is very limited. According to the most-recent review articles, AIE arises because RIM prevents internal motions from 'consuming' excited-state energy, a decidedly Newtonian explanation for a quantum-mechanical process.*"

Indeed, from the macroscopic point of view, we agree with Reviewer 2 that **BDP1** molecules shall be packed in a uniform mode in the ground state. However, things will become different at the molecular or atomic level. As is also well-known, the atoms forming corresponding chemical bonds in organic molecules are constantly vibrating in many different modes no matter existed in solution or solid state, which can be directly reflected by the according infrared spectrum. The atoms will vibrate more drastically when extra energy is pumped in, e.g., laser excitation. Similarly, at excited state, rather than the frozen configuration as what Reviewer 2 has imagined, **BDP1** molecules shall exist in transient configurations with myriad of possibilities that we can't

unambiguously capture at current stage. The transient configurations of **BDP1** molecules would enable the formation of different aggregates that are responsible for the multiple emissions observed in our experimental data (**Figure 1** in the manuscript). Moreover, our quantum mechanical calculation result has clearly provided us the most possible transient configurations of different aggregates that are responsible for the emissions at 605, 768, 868, and 976 nm. Here we would like to add that, the quantum chemical calculation method of “natural transition orbitals (NTOs)” is a commercial software that has been widely used (*J. Am. Chem. Soc.* **139**, 17547-17564 (2017); *J. Am. Chem. Soc.* **133**, 17985-17900 (2011)) in organic luminescence mechanism studies since the year of 1974 (*Int. J. Quantum Chem. Symp.* **8**, 511-513 (1974)).

Furthermore, it is our dream to build an instrument that allows us to capture the transient configurational changes happened at the nanosecond or even shorter lifetime scale at the excited state of **BDP1** solids, so that we will be able to clearly figure out the relationship between the luminescence emission and the packing mode of **BDP1** molecules. However, this would be worth a separate topic that deviates away from the focus of our current manuscript, despite the fact that we will certainly continue working on this “**puzzle**” even without Reviewer 2’s comment.

2. Although the authors made a NTO calculation based on the J-dimer, J-trimer, etc to show different S1 energy level of them to interpret the experimental result, arbitrarily extracting the isolated dimer (or trimer etc) geometry from the single crystal structure is not justified as to represent the whole crystalline bulk.

Response: we want to argue that we have no way of “*arbitrarily extracting the isolated dimer structure*” from a commercial software to purposely satisfy our experimental results. We hope Reviewer 2 can concur on this since we believe that as an expert in solid-state organic luminescent material studies, Reviewer 2 shall be familiar enough with the working principle of the NTO calculation.

3. If the authors want to unambiguously claim that the emitting species are the J-aggregates with different aggregation numbers, the proportion and nature of each species should be uncovered and also why and how they are formed in such a structurally regular single crystal should be rationalized.

Response: This is basically the same question as **Point 1**. Except our above explanation, we have added extra description in our revised manuscript (Page 14), which has clearly stated that “*the formation of different aggregates such as dimer, trimer, etc. are responsible for the emissions observed at 768, 868, and 976 nm, respectively.*” What is more, both the theoretical calculation results (**Figure 5** in Pages 15 and 16) and the detected lifetime variations of the multiple emissions (**Figure 4** in Pages 12-14) can consistently support our explanations.

In summary, rather than the general **common challenges**, we unfortunately don't see any **specific** questions or constructive suggestions from Reviewer 2 in the second round review as other reviewers usually do. For example, Reviewer 1 has provided very helpful suggestions to further improve our manuscript in the first round review. Thus, based on the current comments, we can do nothing in further improving our manuscript and in the meanwhile we feel regretful that this has become the **only and sufficient** reason to not to recommend our manuscript for publication.

Reviewers' Comments:

Reviewer #2:

Remarks to the Author:

In this resubmitted manuscript, I find no substantial answer to my previous question concerning the nature of multiple J-aggregation states in a single crystal. In the rebuttal letter, authors only suppose the time-dependent perturbation or motion-induced restructuring to generate the different aggregation number in a given crystal. If this is actually evidenced in this manuscript, I think it deserved publication in this prestigious journal. However, since it is obviously not the case, I regret I am unable to recommend publication of it.

Reviewer #4:

Remarks to the Author:

This is an interesting paper that shows that the emission of BODIPY derivative is dependent on aggregation state leading to emission in the red and NIR region. The authors did their due diligence to show that the emission is coming from aggregates by conducting precipitation and polymer doping experiments. These experiments show that the emission is evolving as a function of aggregation and concentration. What is strange in the paper is the a similar phenomena is observed in the crystal structure. Aggregate and crystal structure is not the same - one is randomly oriented which can explain the different emissions and one is ordered - and so assuming that different J-aggregates can be addressed in a crystal is a bit mind boggling. I think this explanation is the weakest part of the paper. Also what the authors are calling crystalline is not really clear - did they try to see the emission of a single crystal? Otherwise the effect from microcrystalline powder that might have some defects can explain this discrepancy, i.e., the "crystals" and not really crystalline and behaving as aggregates. In summary I find the paper to be interesting and the results clearly show different emission patterns based on aggregation. This part by itself and associated experimental and theoretical characterization should be enough to be viable for publication in this journal. I would take the crystal part with a grain of salt and maybe try to rewrite it so it will not be such an eye sore.

Response to Reviewers' comments:

All changes mentioned in the response letter have been accordingly red-marked in the revised manuscript and Supporting Information (SI).

Reviewer #2

1. In this resubmitted manuscript, I find no substantial answer to my previous question concerning the nature of multiple J-aggregation states in a single crystal. In the rebuttal letter, authors only suppose the time-dependent perturbation or motion-induced restructuring to generate the different aggregation number in a given crystal. If this is actually evidenced in this manuscript, I think it deserved publication in this prestigious journal. However, since it is obviously not the case, I regret I am unable to recommend publication of it.

Response: We appreciate the reviewer's critical comment. We are sorry that we might have not clearly differentiated the molecular packing order of **BDP** molecules in "*solid state*" from those in "*single crystal state*" that we used for X-ray diffraction studies in previous versions of our manuscript, especially when discussing their fluorescence emission behaviors.

Here we would like to clarify that the fluorescence emissions we have discussed in our manuscript are actually in the *microcrystalline powder state*, NOT the "*single crystal state*" that we have used for X-ray diffraction studies.

We may continue studying the pack orders of **BDP** molecules and corresponding fluorescence emission in the "*single crystal state*", but it has no relationship with this current manuscript and we will report it separately.

To help the readers easily distinguish the existing states of **BDP** molecules in either *solution*, *single crystal*, *microcrystalline powder*, or *thin film*, we have re-described the according expressions in the manuscript and highlighted in red color in the revised manuscript.

Reviewer #4

1. This is an interesting paper that shows that the emission of BODIPY derivative is dependent on aggregation state leading to emission in the red and NIR region. The authors did their due diligence to

show that the emission is coming from aggregates by conducting precipitation and polymer doping experiments. These experiments show that the emission is evolving as a function of aggregation and concentration. What is strange in the paper is the similar phenomena is observed in the crystal structure. Aggregate and crystal structure is not the same - one is randomly oriented which can explain the different emissions and one is ordered - and so assuming that different J-aggregates can be addressed in a crystal is a bit mind boggling. I think this explanation is the weakest part of the paper. Also what the authors are calling crystalline is not really clear - did they try to see the emission of a single crystal? Otherwise the effect from microcrystalline powder that might have some defects can explain this discrepancy, i.e., the "crystals" and not really crystalline and behaving as aggregates. In summary I find the paper to be interesting and the results clearly show different emission patterns based on aggregation. This part by itself and associated experimental and theoretical characterization should be enough to be viable for publication in this journal. I would take the crystal part with a grain of salt and maybe try to rewrite it so it will not be such an eye sore.

Response: Thank you very much for the important comment. Actually what we have meant in our discussion is exactly the *microcrystalline powder* of **BDP** molecules, NOT the *single crystal state*. We have re-described the according expressions in the manuscript so that the authors can easily distinguish the existing states of **BDP** molecules in the state of either *solution*, *single crystal*, *microcrystalline powder*, or *thin film*.

Reviewers' Comments:

Reviewer #4:

Remarks to the Author:

The authors addressed the point that I addressed satisfactorily.

Response to Reviewers' comments:

Reviewer #4

1. The authors addressed the point that I addressed satisfactorily.

Response: Thanks a lot for the critical comment and we really appreciate that.